# Mental health problems and admissions to hospital for accidents and injuries in the UK military: A data linkage study

Zoe Chui[1,4]*, Daniel Leightley[1], Margaret Jones[1], Sabine Landau[2], Paul McCrone[3], Richard D. Hayes[4], Simon Wessely[1,5], Nicola T. Fear[1,5], Laura Goodwin[6]

1 King's Centre for Military Health Research, Institute of Psychiatry, Psychology & Neuroscience, King's College London, London, United Kingdom, 2 Biostatistics & Health Informatics, Institute of Psychiatry, Psychology & Neuroscience, King's College London, London, United Kingdom, 3 Faculty of Education and Health, University of Greenwich, London, United Kingdom, 4 Psychological Medicine, Institute of Psychiatry, Psychology & Neuroscience, King's College London, London, United Kingdom, 5 Academic Department of Military Mental Health, Institute of Psychiatry, Psychology & Neuroscience, King's College London, London, United Kingdom, 6 Spectrum Centre for Mental Health Research, Division of Health Research, Lancaster University, Lancaster, United Kingdom

* zoe.e.chui@kcl.ac.uk

**Data Availability Statement:** The data that support the findings of this study are available from NHS Digital for England, NHS Information Services for Wales and Information Division Services for

## Abstract

### Purpose

Accidents are the most common cause of death among UK military personnel. It is a common misconception in the general public that accidental injuries are always the result of random events, however research suggests that mental health problems and the increased levels of risky behaviour in military personnel may play a role. The objective of this study was to further our understanding of injuries and deaths not related to deployment by examining the associations of mental health, alcohol misuse and smoking with inpatient admission to hospital for accidents and injuries, and attendance to accident and emergency (A&E) departments.

### Methods

Data on all hospital admissions for accidents and injuries and A&E attendance at NHS hospitals in England, Scotland and Wales were linked to data on self-reported mental health problems, alcohol misuse and smoking from a large, representative UK military cohort of serving and ex-serving personnel (n = 8,602). Logistic regression was used to examine the associations between having a hospital admission for an accident or injury with self-reported mental health problems, alcohol misuse and smoking. Cox proportional-hazards regression was then conducted to assess the associations of mental health problems, alcohol misuse and smoking with time to hospital admission for an accident or injury. Finally, negative binomial regression was used to examine associations between the number of A&E attendances with mental health problems, alcohol misuse and smoking.

### Results

Personnel reporting symptoms of common mental disorder (CMD) or probable post-traumatic stress disorder (PTSD) were more likely to have an admission to hospital for an

Scotland for the electronic healthcare records and from King's Centre for Military Health Research for the cohort data, but restrictions apply to the availability of these data, which were used under license for the current study for a limited period only, and so are not publicly available. Data are however available from the Health Research Authority upon reasonable request and with permission of NHS Digital, NHS Information Services for Wales, Information Division Services for Scotland and King's Centre for Military Health Research. A data request can be submitted to the Health Research Authority via contact@hra.nhs.uk quoting reference number 15/CAG/0136.

**Funding:** This work was funded by the Economic and Social Research Council (grant number ES/L014521/1), DL and ZC were funded by the grant. The funder was independent of this research and did not contribute to the design of the study and collection, analysis, interpretation of data and to writing the manuscript. RDH was funded by a Medical Research Council (MRC) Population Health Scientist Fellowship (grant number MR/J01219X/1). RDH and SL have received salary support from the National Institute for Health Research (NIHR) Mental Health Biomedical Research Centre at South London and Maudsley NHS Foundation Trust and King's College London. SW receives funding from the Ministry of Defence. MJ salary is fully supported by a grant from Ministry of Defence. The funders had no role in study design, data collection and analysis, decision to publish, or preparation of the manuscript.

**Competing interests:** RDH has received research funding from Roche, Pfizer, Janssen, Lundbeck and In-Silico-Bioscience. SW at the time of this work was an advisor to the British Army in psychiatry and a trustee of Combat Stress. This work was undertaken prior to NTF's appointment to the Independent Group Advising on the Release of Data. NTF is a trustee of one military related charity. DL is a reservist in the UK Armed Forces. ZC, SL, PM, MJ and LG declare that they have no competing interests. This does not alter our adherence to PLOS ONE policies on sharing data and materials.

accident or injury (fully adjusted odds ratio 1.39, 95% confidence interval [CI] 1.05–1.84), than those who did not report these symptoms, and also had more attendances to A&E (fully adjusted incidence rate ratio [IRR] 1.32, 95% CI 1.16–1.51). A&E attendances were also more common in personnel who were smokers (fully adjusted IRR 1.21, 95% CI 1.09–1.35) following adjustment for demographic, military and health characteristics.

## Conclusions

The findings suggest that accidents and injuries among military personnel are not always random events and that there are health and behavioural factors, including poor mental health and smoking, which are associated (with small effect sizes) with an increased risk of being involved in an accident. Clinicians treating individuals attending hospital after an accident should consider their healthcare needs holistically, including issues related to mental health and health damaging behaviours.

## Introduction

In 2014, the three most common causes of death among serving personnel in the UK Regular Armed Forces were land transport accidents (LTAs; 31%), cancers (22%) and other accidents (18%) [1]. It is a common misperception that accidents are always random, uncontrollable events. For the five-year period 2014–2018, however, UK Regular Armed Forces personnel were at a 66% statistically significant increased risk of death due to an LTA compared to the UK general population, with Army personnel being at a 123% statistically significant increased risk of dying as a result of LTAs [2] and 87% of LTA deaths occurred whilst off duty [2]. These findings suggest that there are differences between military personnel and the general population that may put them at increased risk of accidental death or injury.

There are a number of factors associated with increased risk for accidental death or injury among the general population such as having a mental health problem [3]. Findings from US studies show that military personnel with anxiety and post-traumatic stress disorder (PTSD) have a higher risk of accidental death [4]. UK military personnel self-report a higher prevalence of common mental disorder (CMD) compared to the general population, but there is a lack of research on the relationship between poor mental health and accidental injury [5–7].

Accidental death or injury is more common in individuals who report higher levels of risk-taking behaviour, which may be more likely in military personnel than the general population as an indirect result of individual interests, military training or deployment experiences. For example, having a near-death experience can make people behave more recklessly and have less regard for personal safety [8–10]. Engaging in risky behaviour post-deployment may be a continuation of behavioural patterns that were adaptive in an operational theatre but are less so when personnel return home. Among UK military personnel, deployment is associated with higher levels of risk-taking behaviour such as risky driving [11]. Alternatively, exposure to traumatic events may alter an individual's perception of their risk of being harmed, leading to a personal sense of invulnerability [12]. Exposure to combat traumas are also associated with other externalising behaviours such as post-deployment physical aggression and violence [13], which may involve getting into fights and accidental self-injury. Military personnel may be more likely than the general population to misuse alcohol or other substances to cope with exposure to emotional or physical trauma [14–17], which may in turn increase their risk of accidents and injuries. Whilst this may be a bidirectional relationship among military

personnel, there is evidence in the general population that poor mental health is more likely to result in increased alcohol consumption than vice versa [18].

There is limited research on the prevalence of accidents, which did not result in death, in the UK military. A limitation of focusing on accidental deaths is that it does not tell us whether military personnel are at an increased risk of having an accident per se or whether they have an increased risk of death after having an accident in which they sustained a serious injury. Health behaviours such as alcohol misuse and smoking should also be taken into account, given that drinkers and smokers may have less regard for their health than non-drinkers and non-smokers [19, 20]. Previous work has also shown that pre-military experiences such as adverse family relationships are an important risk factor for ill health in military personnel [21].

The current study addresses this gap in the literature by i) reporting the proportion of hospital admissions for accidents and injuries in a large cohort of UK military personnel; ii) examining the associations between mental health problems (i.e. CMD and PTSD) and health behaviours (i.e. alcohol use and smoking) with a) having a hospital admission for accidents and injuries and b) number of attendances to Accident and Emergency (A&E) departments; iii) assessing the associations of mental health problems, alcohol misuse and smoking with time to hospital admission for an accident or injury. We hypothesised that those reporting mental health problems and poorer health behaviours would be more likely to be admitted to hospital for accident of injury and have a greater number of A&E visits. We also hypothesised time to hospital admission for an accident or injury is more likely to be shorter for those with mental health problems and poor health behaviours compared to those without.

## Methodology

### Study design

A data linkage was conducted between a large UK military cohort study and electronic secondary healthcare records for admissions to hospital in England, Wales and Scotland [22, 23]. The exposures were measured at phase 2 of the cohort study (described below) which took place between 2007 and 2009. Outcomes were reported in the healthcare records starting from the phase 2 questionnaire completion date (November 2007 at the earliest and September 2009 at the latest) until the end of March 2014.

### Data

**Demographic, pre-military, military and health characteristics from the King's Centre for Military Health Research (KCMHR) cohort.** The KCMHR cohort is a large representative study of military personnel. Data were collected in 2004–2006 (phase 1) and again in 2007–2009 (phase 2). Phase 1 recruited approximately 10% of UK military personnel who had been deployed to the first phase of the Iraq war, and a further sample who had not been deployed to Iraq. 10272 participants in total responded (8,686 Regulars, 1,586 Reservists; 59% response rate) [5]. For phase 2 data collection (2007–2009), 9,395 participants from phase 1 were available for follow-up. 6,429 completed the phase 2 data collection (68% response rate). Response at phase 2 was associated with being older, female, an officer and a regular (and so these factors were included in the development of the survey response weights). There were two additional samples at phase 2; with 896 personnel recruited who had deployed to Afghanistan (response rate 50%) and 2,665 individuals responding who had joined the military between April 2003 and April 2007 (response rate 40%). In total, 9,990 individuals completed the phase 2 questionnaire (overall response rate 56%) [6] and 86% of the phase 2 participants provided written informed consent for linkage of their cohort data to healthcare records

(n = 8,602) [22]. Individuals who took part in phase 1 only are not included in the analyses as they had not been asked for consent to access their medical records. The authors obtained section 251 approval of the National Health Service Act 2006 to ensure full compliance with standard ethical procedures.

*Demographics*. Age (at time of phase 2 questionnaire), relationship status and educational attainment (categorised as O-Levels/GCSE or below and A-Levels or higher) were assessed at phase 2. GCSE stands for "General Certificate of Secondary Education" and is an academic qualification in a particular subject, taken in England, Wales and Northern Ireland. Students study for their GCSE's during Year 9 to Year 11, between the ages of 14 and 16. A-Levels (Advanced Levels) are a college or sixth form leaving qualification offered in England, Wales, and Northern Ireland. These are not compulsory, unlike GCSEs, but are generally required across the board for university entrance. A-Levels are taken in Year 12 and 13, between the ages of 16 to 19.

*Pre-military characteristics*. Family relationship adversity in childhood was assessed at phase 2 using a self-report measure consisting of 8 items (e.g. "I used to be hit/hurt by a parent or caregiver regularly") which were summed to create a cumulative measure and analysed as 0–1 and 2+ adversities [21].

*Military characteristics*. Self-reported military rank (other ranks/non-commissioned officer or officer rank), serving status (serving or ex-serving), engagement type (regular or reservist), service branch (Army or Royal Air Force or Naval Services), length of service (less than 4 years [early service leavers] or 4–12 years inclusive or more than 12 years), major operations (no deployment or deployment to Iraq or Afghanistan) and primary role within parent unit (non-combat or combat role) were assessed at phase 2.

*Mental health problems*. Includes symptoms of common mental disorder (CMD) or post-traumatic stress disorder (PTSD) assessed at phase 2. Symptoms of CMD were measured by the General Health Questionnaire– 12, a screening device for identifying minor psychiatric disorders [24, 25]. Examples of items include 'feeling unhappy or depressed' and 'feeling constantly under strain'. Cases were defined as individuals with a score or 4 or more. Symptoms of probable PTSD according to the DSM-IV were assessed by the National Center for PTSD Checklist–Civilian version [26]; a 17-item questionnaire assessing five re-experiencing, seven avoidance and five hyperarousal symptoms, which has previously been used in military populations [6]. Cases were defined as individuals with a total score of 50 or greater and referred to as "probable" PTSD. For all analyses, those reporting symptoms of CMD, and probable PTSD were combined into the category "CMD or probable PTSD" due to low cell sizes.

*Health behaviours*. Alcohol use at phase 2 was measured by the 10-item World Health Organization (WHO) Alcohol Use Disorders Identification Test (AUDIT) [27]. A total AUDIT score of 8 or more was used to define hazardous drinking (a pattern of alcohol consumption carrying with it a risk of harmful consequences to the drinker; these consequences may be damage to health, physical or mental, or they may include social consequences to the drinker or others) [27]. Binge drinking at phase 2 was also assessed using a single item from the AUDIT and defined as drinking six or more units of alcohol on one occasion at least once a week. Smoking status at phase 2 was assessed to distinguish smokers and ex-smokers from non-smokers.

**Admissions to hospital for accidents and injuries and accident and emergency (A&E) attendance from electronic healthcare records (EHRs).** Regulars, reservists and veterans in the UK receive secondary care for physical health conditions from NHS hospitals. Thus, this study combined three NHS datasets from NHS Digital (for Hospital Episode Statistics), the Information Services Division and the Secured Anonymised Information Linkage Databank which contain details of all NHS secondary care in England, Scotland and Wales, respectively.

Unlike many countries, secondary healthcare in England, Scotland and Wales is comprehensively recorded and publicly available through application. Data were requested for the financial years 2003/04 to 2013/2014 in order to cover the timescale of KCMHR cohort study with some follow-up after phase 2. For the purposes of this paper, analyses were restricted to admissions to hospital for accidents and injuries, and number of attendances to accident and emergency (A&E) departments which occurred after phase 2 of the KCMHR cohort study (2007–2009). A dataset of unique patient identifiers including NHS number, forename, surname, sex and date of birth was obtained for each of the three nations. In addition to providing unique regional identifiers, each record and episode was labelled with a unique scrambled cohort identifier to allow for linkage back to the KCMHR military cohort. Details of the data linkage methods are explained in detail elsewhere [22].

*Admissions to hospital for accidents and injuries.* Clinical diagnoses across England, Scotland and Wales were coded using the International Classification of Diseases, 10th revision (ICD-10). NHS Digital advises against employing the full four-character ICD-10 classification code in the EHR, due to possible coding problems [28]. We therefore extracted only the three-character classification which either categorise diagnoses that share common characteristics or represent single conditions. The health outcomes of interest in the current study were accidents and injuries, identified in the healthcare records using the relevant ICD-10 codes (Chapter XX External causes of morbidity and mortality e.g. Accidents V01-X59; Intentional self-harm X60-X84). For the purposes of this paper, only admissions which occurred after phase 2 of the cohort study were included, which was the time that the exposure variables were assessed. The majority of the sample had no admissions to hospital (95.3%), 4.0% had 1 admission and less than 1.0% of the sample had 2 or more admissions to hospital. Thus, a binary variable was created to categorize the number of admissions to hospital as "0" or "1 or more".

*Number of accident and emergency (A&E) attendances.* The number of A&E attendances were counted per individual. Due to large variations in diagnostic coding across the three nations, diagnostic codes were not used and we adopted a rather broad approach to the data by including A&E events for any reason and without restrictions. For the purposes of this paper, only A&E attendances which occurred after phase 2 of the cohort study were included.

## Data analysis

This study included the full sample of those who consented to allow access to their medical records (n = 8602). The analytical sample only included hospital admissions or A&E attendances which occurred after the exposure variables (mental health problem, alcohol misuse and smoking) were collected at phase 2 of the cohort study. Combined sampling weights accounted for both the over-sampling of particular groups (e.g. reservists) at phase 1 and the probability weights of non-response at phase 2. All frequencies were unweighted, and weighted proportions and weighted model estimates were reported using the survey commands in STATA v.15 [29]. A complete case analysis approach was used due to the low proportion of missing data. Missing data on the exposure and potential confounding variables from the KCMHR cohort data (which all 8,602 participants completed) ranged from 0% to 1.1%. Cell sizes less than 8 were not reported according to NHS Digital guidelines [30].

1. **Frequencies of accidents and injuries.** Weighted proportions and 95% confidence intervals were calculated for accidents and injuries, which were ordered by the former. Prevalence of the ten most common accidents and injuries are reported (Table 1).

2. **Demographic and military characteristics of the sample.** Unweighted frequencies and weighted percentages were calculated for a range of demographic and military

**Table 1. Frequencies of accidents and injuries after phase 2 ordered by weighted proportion.**

| Rank | | n | Weighted proportion (95% CI) |
|---|---|---|---|
| 1 | Complications of medical and surgical care (Y40-84) | 96 | 1.19 (0.92–1.47) |
| 2 | Falls (W00-19) | 70 | 0.88 (0.63–1.12) |
| 3 | Exposure to inanimate mechanical forces (W20-49) | 54 | 0.74 (0.51–0.97) |
| 4 | Sequelae of external causes of morbidity and mortality (Y85-89) | 31 | 0.45 (0.27–0.63) |
| 5 | Intentional self-harm (X60-84) | 29 | 0.32 (0.19–0.46) |
| 6 | Accidental exposure to other and unspecified factors (X58-59) | 15 | 0.25 (0.11–0.39) |
| 7 | Assault (X85-Y09) | 25 | 0.24 (0.13–0.34) |
| 8 | Overexertion, travel and privation (X50-57) | 17 | 0.23 (0.10–0.37) |
| 9 | Pedal cyclist injured in transport accident (V10-19) | 20 | 0.21 (0.10–0.32) |
| 10 | Exposure to animate mechanical forces (W50-64) | 15 | 0.16 (0.06–0.25) |

Diagnostic codes from the 10th revision of the International statistical classification of diseases and related health problems (ICD-10) are reported.

characteristics for the full consented cohort and for those with a hospital admission for each of the five most common accidents and injuries separately (Table 2).

3. **Risk factors associated with having an inpatient admission to hospital for an accident or injury.** Logistic regression was conducted to examine associations between the four risk factors (mental health problem, alcohol misuse, binge drinking and smoking status at phase 2) and having an admission for an accident or injury occurring after phase 2 (0 admissions [reference] and 1 or more admissions) (Table 3). The first model adjusted for demographics (age, gender and relationship status) and military characteristics (rank, serving status, engagement type, service branch, major operations and primary role in parent unit). Length of service was excluded from adjusted analyses due to a substantial amount of missing data (3.6%) compared to the other variables (0–1.5%). The second model additionally adjusted for family relationship adversity. Where mental health problem was the exposure variable, the third model additionally adjusted for health behaviours (alcohol misuse, binge drinking and smoking). Odds ratios and 95% confidence intervals are presented for the unadjusted model and the three additionally adjusted models.

4. **Risk factors associated with time to admission to hospital for the five most common accidents and injuries.** Survival analysis was conducted to calculate the time from the date of phase 2 questionnaire completion to discharge for an accident or injury (survival time) (Table 4). Discharge date was used instead of admission date because it more reliably represents one point in time, whereas a person can have multiple admission dates if, for example, they have been transferred from another hospital [31]. To check the assumptions of the Cox proportional-hazards model, tests of proportionality were conducted using the Schoenfeld Residuals Test to examine the association between the residuals for the four risk factors (mental health problem, alcohol misuse, binge drinking and smoking status at phase 2) and time. Tests for each risk factor, as well as a global test, were carried out for the five most common accidents and injuries, and results showed that the proportional-hazards assumption was met. Thus, cox proportional-hazards regression was carried out to assess the effect of the four risk factors on survival time (competing risk analysis was deemed unnecessary as only 40 people in the current sample died of natural causes; 17 had been admitted to hospital for an accident or injury and 22 had been admitted to A&E). Hazard ratios and 95% confidence intervals for the five most common accidents and injuries are

**Table 2. Demographic and military characteristics collected at phase 2 of the full consented cohort and those admitted for the five most common accidents and injuries.**

| | Full consented cohort (n = 8602) | Complications of medical and surgical care (n = 96) | Falls (n = 70) | Exposure to inanimate mechanical forces (n = 54) | Sequelae of external causes of morbidity and mortality (n = 31) | Intentional self-harm (n = 29) |
|---|---|---|---|---|---|---|
| | Total n (weighted column %) | Total n (weighted column %) | Total n (weighted column %) | Total n (weighted column %) | Total n (weighted column %) | Total n (weighted column %) |
| **Age in years (mean, S. D.)** | 35.30 (9.4) | 35.50 (11.1) | 33.94 (9.6) | 33.22 (9.4) | 29.67 (7.4) | 30.41 (9.0) |
| **Relationship status** | | | | | | |
| In a relationship | 6602 (79.1) | 79 (86.7) | 50 (70.8) | 39 (75.5) | 23 (74.7) | 21 (74.6) |
| Not in a relationship | 1973 (20.9) | 16 (14.3) | 20 (29.2) | 14 (24.5) | 8 (25.3) | 8 (25.4) |
| **Educational attainment** | | | | | | |
| GCSE (O-Level) or below | 3517 (44.7) | 38 (38.9) | 30 (43.8) | 26 (45.1) | 15 (49.1) | (*) |
| A-level or higher | 4730 (55.3) | 55 (61.1) | 37 (56.2) | 24 (54.9) | 16 (50.9) | (*) |
| **Family relationship adversity in childhood** | | | | | | |
| 0–1 factors | 5496 (64.4) | 47 (46.5) | 39 (60.1) | 32 (60.8) | 17 (60.5) | 15 (50.8) |
| 2+ factors | 3031 (35.6) | 49 (53.5) | 31 (39.9) | 22 (39.2) | 13 (39.5) | 14 (49.2) |
| **Military rank** | | | | | | |
| Officers | 1930 (20.5) | 21 (18.8) | 13 (14.9) | (*) | (*) | (*) |
| NCOs/Other Ranks | 6672 (79.5) | 75 (81.2) | 57 (85.1) | (*) | (*) | (*) |
| **Serving status** | | | | | | |
| Serving | 6471 (71.3) | 75 (73.4) | 46 (59.8) | 38 (60.6) | (*) | 18 (46.0) |
| Ex-serving | 2117 (28.7) | 21 (26.6) | 24 (40.2) | 16 (39.4) | (*) | 11 (54.0) |
| **Engagement type** | | | | | | |
| Regular | 7059 (88.7) | 73 (82.9) | 57 (91.0) | 46 (89.8) | (*) | (*) |
| Reserve | 1543 (11.3) | 23 (17.1) | 13 (9.0) | 8 (10.2) | (*) | (*) |
| **Service branch** | | | | | | |
| Army | 5628 (71.2) | 71 (75.7) | 50 (70.3) | (*) | (*) | (*) |
| RAF | 1657 (14.7) | 9 (9.5) | 11 (17.1) | (*) | (*) | (*) |
| Naval Services | 1317 (14.1) | 16 (14.8) | 9 (12.6) | (*) | (*) | (*) |
| **Length of service** | | | | | | |
| Less than 4 years | 588 (4.8) | 10 (6.8) | (*) | (*) | (*) | (*) |
| 4–12 years inclusive | 3675 (41.0) | 40 (43.5) | (*) | (*) | (*) | (*) |
| More than 12 years | 4032 (54.2) | 42 (49.7) | (*) | (*) | (*) | (*) |
| **Major operations** | | | | | | |
| No deployment | 4423 (54.8) | 30 (37.3) | 23 (45.9) | 19 (47.0) | 10 (35.8) | 10 (48.9) |
| Deployment to Iraq or Afghanistan | 4179 (45.2) | 66 (62.7) | 47 (54.1) | 35 (53.0) | 21 (64.2) | 19 (51.1) |
| **Primary role within parent unit** | | | | | | |
| Non-combat role | 6477 (76.0) | 70 (72.1) | 49 (69.4) | 37 (67.5) | 17 (55.2) | 21 (69.2) |
| Combat role | 2035 (24.0) | 25 (27.9) | 20 (30.6) | 16 (32.5) | 14 (44.8) | 8 (30.8) |
| **CMD or probable PTSD** | | | | | | |
| Non-case | 6828 (79.5) | 67 (75.2) | 58 (83.2) | 40 (69.8) | 21 (64.5) | 16 (59.0) |
| Case | 1734 (20.5) | 27 (24.8) | 11 (16.8) | 14 (30.2) | 10 (35.5) | 13 (41.0) |
| **Alcohol misuse** | | | | | | |

*(Continued)*

**Table 2.** (Continued)

| | Full consented cohort (n = 8602) | Complications of medical and surgical care (n = 96) | Falls (n = 70) | Exposure to inanimate mechanical forces (n = 54) | Sequelae of external causes of morbidity and mortality (n = 31) | Intentional self-harm (n = 29) |
|---|---|---|---|---|---|---|
| Non-case | 3670 (43.5) | 37 (41.2) | 24 (32.0) | 21 (39.1) | (*) | 13 (34.4) |
| Case (AUDIT 8+) | 4827 (56.5) | 58 (58.8) | 44 (68.0) | 33 (60.9) | (*) | 16 (65.6) |
| **Binge drinking** | | | | | | |
| Monthly or less | 5216 (61.6) | 58 (62.8) | 37 (51.9) | 28 (54.9) | 12 (39.2) | 21 (64.8) |
| Weekly or more | 3266 (38.4) | 37 (37.2) | 30 (48.1) | 26 (45.1) | 18 (60.8) | 8 (35.2) |
| **Smoking status** | | | | | | |
| Non-smoker | 4293 (48.7) | 42 (44.5) | 40 (52.0) | 27 (53.9) | 11 (42.6) | 10 (27.7) |
| Smoker or Ex-smoker | 4193 (51.3) | 53 (55.5) | 29 (48.0) | 27 (46.1) | 19 (57.4) | 19 (72.3) |

Unweighted frequencies and weighted percentages are reported. Numbers might not add up to totals because of missing data. (*) = Numbers and percentages are not reported when n is smaller than 8 due to Hospital Episode Statistics reporting guidelines. NCO = Non-commissioned officer. RAF = Royal Air Force. Naval services = Royal Navy and Royal Marines.

**Table 3. Risk factors at phase 2 associated with having an admission to hospital for an accident or injury.**

| | 0 admissions | 1+ admission | | | | |
|---|---|---|---|---|---|---|
| | Total n (weighted column %) | Total n (weighted column %) | Unadjusted OR (95% CI) | Adjusted OR (95% CI)[1] | Adjusted OR (95% CI)[2] | Adjusted OR (95% CI)[3] |
| **CMD or probable PTSD** | | | | | | |
| Non-case | 6,541 (79.9) | 287 (72.5) | 1.00 | 1.00 | 1.00 | 1.00 |
| Case | 1,625 (20.1) | 109 (27.5) | 1.54 (1.18–2.01) * | 1.45 (1.10–1.89) * | 1.38 (1.04–1.82) * | 1.39 (1.05–1.84) * |
| **Alcohol misuse** | | | | | | |
| Non-case | 3,516 (43.8) | 154 (38.6) | 1.00 | 1.00 | 1.00 | 1.00 |
| Case (AUDIT 8+) | 4,586 (56.2) | 241 (61.4) | 1.24 (0.97–1.58) | 1.11 (0.86–1.43) | 1.08 (0.84–1.40) | 1.06 (0.82–1.37) |
| **Binge drinking at phase 2** | | | | | | |
| Monthly or less | 4,981 (61.7) | 235 (58.6) | 1.00 | 1.00 | 1.00 | 1.00 |
| Weekly or more | 3,107 (38.3) | 159 (41.4) | 1.14 (0.89–1.45) | 1.05 (0.82–1.34) | 1.03 (0.81–1.32) | 1.01 (0.79–1.29) |
| **Smoking status at phase 2** | | | | | | |
| Non-smoker | 4,109 (48.9) | 184 (44.3) | 1.00 | 1.00 | 1.00 | 1.00 |
| Smoker or Ex-smoker | 3,982 (51.1) | 211 (55.7) | 1.20 (0.95–1.53) | 1.11 (0.87–1.42) | 1.10 (0.86–1.40) | 1.09 (0.85–1.39) |

*p < .05. OR = odds ratio. CI = confidence interval.

Unweighted frequencies and weighted percentages are reported.

[1] Adjusted for age, relationship status and military characteristics (rank, serving status, engagement type, service branch, major operations and primary role in parent unit)

[2] Additional adjustment for family relationship adversity in childhood

[3] Additional adjustment for alcohol misuse and/or smoking status

**Table 4. Risk factors at phase 2 associated with time to admission to hospital for the five most common accidents and injuries.**

| | Unadjusted HR (95% CI) | Adjusted HR (95% CI)[1] | Adjusted HR (95% CI)[2] | Adjusted HR (95% CI)[3] |
|---|---|---|---|---|
| Complications of medical and surgical care (n = 96) | | | | |
| **CMD or probable PTSD** | n = 8,562 | n = 8,444 | n = 8,364 | n = 8,232 |
| Non-case | 1.00 | 1.00 | 1.00 | 1.00 |
| Case | 1.30 (0.76–2.21) | 1.32 (0.76–2.27) | 1.20 (0.69–2.10) | 1.21 (0.68–2.14) |
| **Alcohol misuse** | n = 8,497 | n = 8,042 | n = 7,973 | |
| Non-case | 1.00 | 1.00 | 1.00 | - |
| Case (AUDIT 8+) | 1.13 (0.70–1.84) | 1.14 (0.66–1.95) | 1.08 (0.63–1.85) | - |
| **Binge drinking** | n = 8,482 | n = 8,369 | n = 8,297 | |
| Monthly or less | 1.00 | 1.00 | 1.00 | - |
| Weekly or more | 0.97 (0.60–1.57) | 0.97 (0.59–1.60) | 0.94 (0.57–1.54) | - |
| **Smoking status** | n = 8,486 | n = 8,370 | n = 8,315 | |
| Non-smoker | 1.00 | 1.00 | 1.00 | - |
| Smoker or Ex-smoker | 1.19 (0.74–1.90) | 1.12 (0.69–1.82) | 1.08 (0.67–1.75) | - |
| Falls (n = 70) | | | | |
| **CMD or probable PTSD** | n = 8,562 | n = 8,444 | n = 8,364 | n = 8,232 |
| Non-case | 1.00 | 1.00 | 1.00 | 1.00 |
| Case | 0.78 (0.35–1.70) | 0.69 (0.31–1.53) | 0.68 (0.30–1.54) | 0.64 (0.28–1.50) |
| **Alcohol misuse** | n = 8,497 | n = 8,383 | n = 8,311 | |
| Non-case | 1.00 | 1.00 | 1.00 | - |
| Case (AUDIT 8+) | 1.64 (0.91–2.93) | 1.40 (0.81–2.43) | 1.39 (0.79–2.43) | - |
| **Binge drinking** | n = 8,482 | n = 8,369 | n = 8,297 | |
| Monthly or less | 1.00 | 1.00 | 1.00 | - |
| Weekly or more | 1.49 (0.84–2.63) | 1.37 (0.80–2.34) | 1.36 (0.79–2.34) | - |
| **Smoking status** | n = 8,486 | n = 8,370 | n = 8,315 | |
| Non-smoker | 1.00 | 1.00 | 1.00 | - |
| Smoker or Ex-smoker | 0.88 (0.50–1.55) | 0.84 (0.47–1.51) | 0.83 (0.46–1.51) | - |
| Exposure to inanimate mechanical forces (n = 54) | | | | |
| **CMD or probable PTSD** | n = 8,562 | n = 8,444 | n = 8,364 | n = 8,232 |
| Non-case | 1.00 | 1.00 | 1.00 | 1.00 |
| Case | 1.67 (0.82–3.37) | 1.57 (0.79–3.13) | 1.55 (0.77–3.14) | 1.56 (0.78–3.15) |
| **Alcohol misuse** | n = 8,497 | n = 8,383 | n = 8,311 | |
| Non-case | 1.00 | 1.00 | 1.00 | - |
| Case (AUDIT 8+) | 1.18 (0.62–2.24) | 1.11 (0.55–2.25) | 1.09 (0.54–2.21) | - |
| **Binge drinking** | n = 8,482 | n = 8,369 | n = 8,297 | |
| Monthly or less | 1.00 | 1.00 | 1.00 | - |
| Weekly or more | 1.31 (0.70–2.45) | 1.20 (0.63–2.28) | 1.19 (0.62–2.27) | - |
| **Smoking status** | n = 8,486 | n = 8,370 | n = 8,315 | |
| Non-smoker | 1.00 | 1.00 | 1.00 | - |
| Smoker or Ex-smoker | 0.82 (0.44–1.52) | 0.77 (0.40–1.46) | 0.76 (0.40–1.44) | - |
| Sequelae of external causes of morbidity and mortality (n = 31) | | | | |
| **CMD or probable PTSD** | n = 8,562 | n = 8,454 | n = 8,376 | n = 8,242 |
| Non-case | 1.00 | 1.00 | 1.00 | 1.00 |
| Case | 2.16 (0.92–5.08) | 1.98 (0.85–4.62) | 1.87 (0.71–4.91) | 1.80 (0.68–4.74) |
| **Alcohol misuse** | n = 8,497 | n = 8,393 | n = 8,321 | |
| Non-case | 1.00 | 1.00 | 1.00 | - |
| Case (AUDIT 8+) | 2.59 (1.02–6.55) * | 1.94 (0.78–4.84) | 1.93 (0.77–4.80) | - |

(*Continued*)

**Table 4.** (Continued)

| | Unadjusted HR (95% CI) | Adjusted HR (95% CI)[1] | Adjusted HR (95% CI)[2] | Adjusted HR (95% CI)[3] |
|---|---|---|---|---|
| **Binge drinking** | n = 8,482 | n = 8,040 | n = 8,307 | |
| Monthly or less | 1.00 | 1.00 | 1.00 | - |
| Weekly or more | 2.55 (1.13–5.76) * | 2.10 (0.98–4.53) | 2.10 (0.97–4.54) | - |
| **Smoking status** | n = 8,486 | n = 8,380 | n = 8,325 | |
| Non-smoker | 1.00 | 1.00 | 1.00 | - |
| Smoker or Ex-smoker | 1.29 (0.56–2.97) | 1.21 (0.52–2.82) | 1.21 (0.52–2.83) | - |
| Intentional self-harm (n = 29) | | | | |
| **CMD or probable PTSD** | n = 8,562 | n = 8,444 | n = 8,364 | n = 8,232 |
| Non-case | 1.00 | 1.00 | 1.00 | 1.00 |
| Case | 2.69 (1.16–6.26) * | 2.28 (0.92–5.64) | 2.13 (0.83–5.42) | 2.08 (0.80–5.41) |
| **Alcohol misuse** | n = 8,497 | n = 8,383 | n = 8,311 | |
| Non-case | 1.00 | 1.00 | 1.00 | - |
| Case (AUDIT 8+) | 1.50 (0.67–3.35) | 1.23 (0.53–2.84) | 1.18 (0.51–2.72) | - |
| **Binge drinking** | n = 8,482 | n = 8,369 | n = 8,297 | |
| Monthly or less | 1.00 | 1.00 | 1.00 | - |
| Weekly or more | 0.88 (0.34–2.31) | 0.77 (0.30–1.93) | 0.74 (0.30–1.85) | - |
| **Smoking status** | n = 8,486 | n = 8,370 | n = 8,315 | |
| Non-smoker | 1.00 | 1.00 | 1.00 | - |
| Smoker or Ex-smoker | 2.48 (1.04–5.95) * | 2.37 (1.01–5.60)* | 2.30 (0.98–5.36) | - |

*p < .05. HR = hazard ratio. CI = confidence interval. (*) = Numbers and percentages are not reported when n is smaller than 8 due to Hospital Episode Statistics reporting guidelines.

[1]Adjusted for age, relationship status and military characteristics where cell sizes are 8 or above (see below)

- *Complications of medical and surgical care*: age, relationship status, military rank, serving status, engagement type, service branch, major operations and primary role within parent unit

- *Falls*: age, relationship status, military rank, serving status, engagement type, service branch, major operations and primary role within parent unit

- *Exposure to inanimate mechanical forces*: age, relationship status, serving status, engagement type, major operations and primary role within parent unit

- *Sequelae of external causes of morbidity and mortality*: age, relationship status, major operations and primary role within parent unit

- *Intentional self-harm*: age, relationship status, serving status, major operations and primary role within parent unit

[2]Additional adjustment for family relationship adversity in childhood

[3]Additional adjustment for alcohol misuse and smoking status

presented. The first model adjusted for demographic factors (age, gender, and relationship status) and military characteristics (rank, serving status, engagement type, service branch, major operations and primary role in parent unit [military characteristics with cell sizes of less than 8 were excluded from adjusted analyses in accordance with NHS Digital's publication policy on small numbers]) and the second model additionally adjusted for family relationship adversity. Where mental health problem was the exposure variable, the third model additionally adjusted for health behaviours (alcohol misuse, binge drinking and smoking).

5. **Risk factors associated with the number of attendances to Accident and Emergency (A&E) departments.** Count data were tested for positive skew, over-dispersion and independence of events to determine which regression analysis technique to use. Once chi-squared goodness-of-fit tests established that data did not follow a Poisson distribution, negative binomial regression was conducted to examine associations between four risk factors (mental health problem, alcohol misuse, binge drinking and smoking status at phase 2) and the number of A&E attendances (Table 5). Models were adjusted as described previously.

**Table 5. Risk factors at phase 2 associated with the number of subsequent attendances to accident and emergency (A&E) departments.**

| | Median number of A&E attendances (IQR)[1] | Unadjusted IRR (95% CI) | Adjusted IRR (95% CI)[2] | Adjusted IRR (95% CI)[3] | Adjusted IRR (95% CI)[4] |
|---|---|---|---|---|---|
| **CMD or probable PTSD** | | | | | |
| Non-case | 1 (1–3) | 1.00 | 1.00 | 1.00 | 1.00 |
| Case | 2 (1–3) | 1.47 (1.28–1.69) * | 1.32 (1.17–1.50) * | 1.32 (1.16–1.50) * | 1.32 (1.16–1.51) * |
| **Alcohol misuse** | | | | | |
| Non-case | 1 (1–2) | 1.00 | 1.00 | 1.00 | - |
| Case (AUDIT 8+) | 2 (1–3) | 1.26 (1.12–1.42) * | 1.08 (0.96–1.20) | 1.07 (0.95–1.19) | - |
| **Binge drinking** | | | | | |
| Monthly or less | 1 (1–3) | 1.00 | 1.00 | 1.00 | - |
| Weekly or more | 2 (1–3) | 1.09 (0.97–1.23) | 0.99 (0.89–1.10) | 0.98 (0.88–1.10) | - |
| **Smoking status** | | | | | |
| Non-smoker | 1 (1–2) | 1.00 | 1.00 | 1.00 | - |
| Smoker or Ex-smoker | 2 (1–3) | 1.31 (1.17–1.47) * | 1.22 (1.10–1.35) * | 1.21 (1.09–1.35) * | - |

*p < .05. IQR = interquartile range. IRR = incidence rate ratio. CI = confidence interval.

[1]Weighted medians and interquartile ranges are reported

[2]Adjusted for age, relationship status and military characteristics (rank, serving status, engagement type, service branch, major operations and primary role in parent unit)

[3]Additional adjustment for family relationship adversity in childhood

[4]Additional adjustment for alcohol misuse and/or smoking status

## Ethics statement

This study received ethical approval from the London-Dulwich NHS Research Ethics Committee in November 2014 (REC no: 07/Q0703/36) and Section 251 approval from the Health Research Authority (Ref: 15/CAG/0136). Written informed consent was obtained from all participants for inclusion in the study.

## Results

### Prevalence of accidents and injuries

The proportion of hospital admissions for accidents and injuries in this sample was low (5.3%). The prevalence of the five most common reasons for an admission to hospital ranged from 1.2% (most common) to 0.3% (least common). In order of most to least common, the first was complications of medical and surgical care (Y40-Y84), two-thirds of which involved the implant of an artificial internal device or removal of an organ. The second was falls (W00-W19), which mainly consisted of falls on the same level due to a collision with, or pushing by, another person. The third was exposure to inanimate mechanical forces (W20-W49), which involved being crushed or jammed between objects, or coming into contact with sharp glass (Table 1). The fourth most common reason for admission was sequelae of external causes of morbidity and mortality (Y85-Y89), that is, late effects of a previous injury requiring hospital admission which, in this sample, were typically the result of transport accidents or "unspecified" accidents. Finally, the fifth most common reason was intentional self-harm (X60-X84), with the majority of admissions involving intentional self-poisoning by analgesics, antiepileptic and psychotropic drugs, narcotics, hallucinogens and alcohol. Results of other reasons for admission for the five most prevalent accidents and injuries in our sample are not reported due to small numbers.

## Demographic and military characteristics of those admitted for the five most common accidents and injuries

Table 2 presents the demographic and military characteristics of the full consented cohort and those with a hospital admission for the five most common accidents and injuries. Admissions for complications of medical and surgical care were more common among those who reported family relationship adversity in childhood and less common in those serving in the Royal Air Force (RAF) compared to those serving in the Army. Admissions for sequelae of external causes of morbidity and mortality were more common among those who had served in a combat role and those reporting binge drinking. Admissions for intentional self-harm were more common among ex-serving personnel, smokers and those with mental health problems compared to serving personnel, non-smokers and those without a mental health problem, respectively.

## Risk factors associated with having an inpatient admission to hospital for accidents and injuries

In the unadjusted model, those with mental health problems were 1.5 times more likely than those without a mental health problem to have an admission to hospital for accidents and injuries (Table 3). This effect remained significant in all adjusted models. Alcohol misuse, binge drinking, and smoking were not significantly associated with having an admission to hospital for accidents and injuries.

## Risk factors for time to admission to hospital for accidents and injuries

**Complications of medical and surgical care, falls and exposure to inanimate mechanical forces.** We found no evidence of associations between mental health and health behaviours with time to admission for complications of medical and surgical care, falls and exposure to inanimate mechanical forces (Table 4).

**Sequelae of external causes of morbidity and mortality.** Those who reported alcohol misuse or binge drinking had over 2.5 times the hazard of an admission to hospital for sequelae of external causes of morbidity and mortality compared to those who didn't engage in these drinking behaviours, however these effects diminished to non-significance following adjustment for demographic and military characteristics (Table 4).

**Intentional self-harm.** Those who reported having mental health problems and current smokers had 2.7 and 2.5 times the hazard, respectively, of admission to hospital for intentional self-harm compared to those without mental health problems and non-smokers, however these effects diminished to non-significance following adjustment for demographic and military characteristics (Table 4).

## Risk factors associated with the number of attendances to A&E

In the full sample, the median number of A&E attendances was 1 (IQR 1–3). For those who reported mental health problems, alcohol misuse, binge drinking and smoking, the median number of A&E attendances was 2 (IQR 1–3) (Table 5). Those with mental health problems were 1.3 times more likely than those without mental health problems to have a greater number of A&E attendance in the fully adjusted model. Smokers or ex-smokers were 1.2 times more likely than non-smokers to have a greater number of A&E attendances in the fully adjusted model. Those who reported alcohol misuse (i.e. an AUDIT score of 8 or more) were 1.3 times more likely than those who didn't misuse alcohol to have a greater number of

attendances to A&E in the unadjusted model, however this association was attenuated following adjustment for demographic and military characteristics.

## Discussion

### Key findings

The main findings of the current study show that, among UK military personnel, poor mental health and smoking are associated (with small effect sizes) with an increased risk of being admitted to hospital for an accident or injury. Those with symptoms of probable PTSD or CMD were significantly more likely to have an admission to hospital for an accident or injury, and were more likely to have a greater number of attendances to A&E. A&E attendance was also more likely in smokers. Our results suggest that within a population of serving and ex-serving military personnel there are individual level factors that increase the likelihood of accidents occurring which require hospital treatment, and by providing appropriate mental health treatments we may be able to reduce these risks.

### Risk factors associated with admission to hospital for accidents and injuries and A&E attendance

Military personnel are often seen as less vulnerable to physical health problems than the general population due to their high levels of fitness. However, the current study has identified factors which are associated with an increased likelihood of accidents and injuries in military personnel, such as mental health problems. Although mental disorders are well-known risk factors for suicide, little attention has been given to their impact on accidental injury and death. Previous research has suggested that accidental death shares risk factors with suicide and premature death [32]. Indifference to death, in contrast with a desire to die, may lead to an increased risk of accidental injury or death, which cannot be addressed through studies or interventions that treat accidents as random events and focus on suicide [33, 34]. According to sensation seeking theory, high sensation seekers are well suited for warlike situations and would be welcomed in special high-risk operations [35]. However, it is possible that these traits may lead to unnecessary or non-relevant forms of risk-taking, such as alcohol use or dangerous driving, once personnel return from deployment. These findings potentially have implications for the prevention of premature death among people with mental illness.

Another risk factor for premature death is smoking [36] and our findings suggest that smoking is associated with accidents and injuries among military personnel (with small effect sizes). Smokers are more likely than non-smokers to have chronic health conditions such as diabetes, cancer and obesity [37–39] and are also more likely to engage in behaviours that are damaging to oral health [40]. Based on evidence from previous research, we hypothesise that this combination of having poorer health and also engaging in health damaging behaviours may make smokers more likely to be involved in accidents. Our findings support previous studies showing that smoking can increase the risk of experiencing an accidental injury and/or result in poorer outcomes after an injury. For example, research has shown that smokers are more likely to experience stress fractures, and take longer to heal than non-smokers [41, 42]. A previous study of military personnel from the UK found that deployment is associated with risky driving (i.e. speeding or driving without a seatbelt), and that risk-taking behaviours covary as demonstrated by the clear association between risky driving, heavy smoking and heavy alcohol use [43]. While taking risks may be more likely during deployment, it is important that military personnel returning to civilian life do not continue to engage in risk-taking behaviours that could result in accidental injury or taking longer to recover from injuries.

The five most common reasons for admission for an accident or injury in this population were complications of medical and surgical care, falls, exposure to inanimate mechanical forces, sequelae of external causes of morbidity and mortality and intentional self-harm. The findings showed that having a mental health problem and smoking were only associated with intentional self-harm in unadjusted models but effects diminished after accounting for demographic and military characteristics so may be explained by other factors including age, relationship status, military characteristics or family relationship adversity [21, 44–46].

## Strengths and limitations

The strengths of this study are that it is to our knowledge the first study to analyse data on accidents and injuries from a unique dataset which linked a large, representative UK military cohort to electronic healthcare records for England, Scotland and Wales. This provided a UK-wide medical history for 8,602 military personnel. Due to the quality and completeness of A&E data, however, only the number of attendances were included in the analyses and information about the reason for attendance could not be retrieved. Furthermore, some of the associations should be treated with caution due to small cell sizes, which is a limitation of the rarity of some outcomes and the size of the cohort available for the study. As a result, some analyses produced only small effect sizes and small numbers may have increased the risk of type II error, also known as a false negative. In terms of examining associations with mental health problems, the authors recognise that CMD and PTSD are different conditions but had to be combined due to small cell sizes. As the first data linkage of this kind, we have published all outcomes relevant to the aims of this study whilst ensuring that reporting guidelines for administrative data were adhered to and individuals were not identifiable from small cell sizes. In some cases, diagnostic codes from the hospital admission data did not provide sufficient detail to draw conclusions about the impact of poor mental health and smoking on an individual's risk of having an accident. Whilst coding systems used in U.S. based military hospitals, such as the NATO Standardized Agreement (STANAG) 2050, may provide more detail about the nature of the injury, this study provides a better picture of injury events both in and out of military settings and additionally for those who have already left the military. For this paper, hospital admissions were identified in healthcare records using the internationally recognised ICD-10 codes in Chapter XX "External causes of morbidity and mortality". We acknowledge that this ICD-10 Chapter includes a range of differing outcomes. Finally, the current study used the full consented sample (n = 8,602) as the denominator for analyses, rather than only those who were matched (n = 6,336). Efforts were taken to reduce potential sources of bias by comparing any differences between the matched (n = 6,336) and unmatched (n = 8,602) samples; differences were very minimal and have been reported previously [23]. The greatest difference between these samples was by presence of NHS number, which would be expected given this was used in the matching process. We therefore took what was felt to be the conservative option in treating those who were not matched as not having a hospital episode, given that those who required NHS treatment were more likely to have an NHS number and therefore have been matched.

## Implications

A more comprehensive understanding of the relationships between mental disorders and accidental death is needed to enable better risk management. The current drive to reduce the 10-year mortality gap in those with common mental disorders focuses predominantly on suicide prevention, provision of care for substance misuse issues and incentive schemes for GPs to encourage monitoring of physical health [47]. Accidents tend to be overlooked in these

recommendations and may help to address the significant health inequalities for those living with common mental disorders.

## Conclusions

Military personnel who smoke or have mental health problems may have a greater risk of having an accident or injury compared to those without these problems. This potentially calls for targeted interventions aimed at such individuals to treat mental health problems and, crucially, promote awareness of personal safety and health. This is especially important for service leavers who may consider civilian occupations to be relatively safe compared to their previous role in the military. For military populations in the UK, more attention needs to be paid to the risk and prevention of accidental injury and death.

## Author Contributions

**Conceptualization:** Laura Goodwin.

**Formal analysis:** Zoe Chui.

**Funding acquisition:** Laura Goodwin.

**Methodology:** Daniel Leightley, Laura Goodwin.

**Supervision:** Laura Goodwin.

**Writing – original draft:** Zoe Chui.

**Writing – review & editing:** Zoe Chui, Daniel Leightley, Margaret Jones, Sabine Landau, Paul McCrone, Richard D. Hayes, Simon Wessely, Nicola T. Fear, Laura Goodwin.

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
