## [Decision Letter · Decision Letter 0]

18 May 2022

PONE-D-21-32763Using electronic hospital records to identify risk factors for accidents and injuries in the UK military: adata linkage studyPLOS ONE

Dear Dr. Chui,

Thank you for submitting your manuscript to PLOS ONE. After careful consideration, we feel that it has merit but does not fully meet PLOS ONE’s publication criteria as it currently stands. Therefore, we invite you to submit a revised version of the manuscript that addresses the points raised during the review process.

Please note that we have only been able to secure a single reviewer to assess your manuscript. We are issuing a decision on your manuscript at this point to prevent further delays in the evaluation of your manuscript. Please be aware that the editor who handles your revised manuscript might find it necessary to invite additional reviewers to assess this work once the revised manuscript is submitted. However, we will aim to proceed on the basis of this single review if possible.

The reviewer has raised a number of concerns that need attention. They request additional information on methodological aspects of the study, revisions to the statistical analyses and they question the internal and external validity of the results reported.

Could you please revise the manuscript to carefully address the concerns raised?

We look forward to receiving your revised manuscript.

Kind regards,

Sebastian Shepherd

Staff Editor

PLOS ONE

Journal Requirements:

This work was funded by the Economic and Social Research Council (grant number ES/L014521/1), DL and ZC were funded by the grant. The funder was independent of this research and did not contribute to the design of the study and collection, analysis, interpretation of data and to writing the manuscript. RDH was funded by a Medical Research Council (MRC) Population Health Scientist Fellowship (grant number MR/J01219X/1). RDH and SL have received salary support from the National Institute for Health Research (NIHR) Mental Health Biomedical Research Centre at South London and Maudsley NHS Foundation Trust and King’s College London. SW receives funding from the Ministry of Defence. MJ salary is fully supported by a grant from Ministry of Defence.

5. Thank you for stating the following in the Acknowledgments/Funding Section of your manuscript: 

RDH has received salary support from the National Institute for Health Research (NIHR) Biomedical Research Centre at South London and Maudsley NHS Foundation Trust and King's College London.

This work was funded by the Economic and Social Research Council (grant number ES/L014521/1), DL and ZC were funded by the grant. The funder was independent of this research and did not contribute to the design of the study and collection, analysis, interpretation of data and to writing the manuscript. RDH was funded by a Medical Research Council (MRC) Population Health Scientist Fellowship (grant number MR/J01219X/1). RDH and SL have received salary support from the National Institute for Health Research (NIHR) Mental Health Biomedical Research Centre at South London and Maudsley NHS Foundation Trust and King’s College London. SW receives funding from the Ministry of Defence. MJ salary is fully supported by a grant from Ministry of Defence.

6. Thank you for stating the following in the Competing Interests section: 

RDH has received research funding from Roche, Pfizer, Janssen and Lundbeck. NTF is a trustee of a veteran’s charity and a specialist member of the independent group advising NHS Digital about the release of patient data. 

7. We note that you have stated that you will provide repository information for your data at acceptance. Should your manuscript be accepted for publication, we will hold it until you provide the relevant accession numbers or DOIs necessary to access your data. If you wish to make changes to your Data Availability statement, please describe these changes in your cover letter and we will update your Data Availability statement to reflect the information you provide.

8. Your ethics statement should only appear in the Methods section of your manuscript. If your ethics statement is written in any section besides the Methods, please move it to the Methods section and delete it from any other section. Please ensure that your ethics statement is included in your manuscript, as the ethics statement entered into the online submission form will not be published alongside your manuscript. 

Reviewers' comments:

Reviewer's Responses to Questions

**Comments to the Author**

1. Is the manuscript technically sound, and do the data support the conclusions?

Reviewer #1: Partly

2. Has the statistical analysis been performed appropriately and rigorously? 

Reviewer #1: Yes

3. Have the authors made all data underlying the findings in their manuscript fully available?

Reviewer #1: Yes

4. Is the manuscript presented in an intelligible fashion and written in standard English?

Reviewer #1: Yes

5. Review Comments to the Author

Reviewer #1: The authors report some results from the King’s Centre for Military Health Research cohort. The investigators used data from a prospective survey and health records to examine injuries and accidents. They found that PTSD or a common mental health disorder were associated with hospitalization for an injury or accident and more accident and emergency (A&E) department visits. A&E department visits were also associated with smoking.

This is a complex and ambitious study that has collected data from thousands of military members over many years. The fact that the survey results were linked to electronic health records is important and adds to the value of the results.

However, I have a several recommendations to improve the manuscript:

1. The title does not seem descriptive enough to me. The current title reads more like a methods paper to me. I do not see the data linkage and use of EHRs as the key aspect of the findings. This paper seems to be reporting how mental health and military characteristics are associated with a variety of injury and accident outcomes; I recommend working some version of that into the title.

2. Lines 97-99 seem too strong regarding deployment. There are many possibilities for why military personnel may have more injuries and higher risk taking than the general population. (Many of these are mentioned later in the Discussion). The military may enlist people with higher risk-taking due to their individual interests or the military’s recruitment strategies. Other military experiences and training may increase risk taking. Deployment is not the first thing that comes to mind. I recommend acknowledging several possibilities and then going on to discuss deployment as one possibility.

3. I recommend re-naming “any mental health problem” to something like “Either CMD or PTSD,” since the authors did not assess for any mental health problem.

4. I was fairly late into the paper before I realized that self-inflicted injuries (i.e. suicidal behaviors) were included. Given the set-up of the paper, this surprised me, and I do not typically think of self-inflicted injuries in the same category as some of the accidents and other injuries. Is it possible that different results would be found with more specific categories?

5. There are so few accidents and injuries for most categories… I do not think there is anything the authors can do about this, but this reduces the impact of the paper.

6. Table 2 does not appear to indicate statistical significance as reported in the results (and the other tables).

7. Binge drinking results are reported for Table 2 in the Results section but I did not see it in Table 2.

8. Table 2: For the 4th and 5th most common accidents and injuries (columns 4 and 5), the authors suppressed almost half the variables (5/12) due to low n’s. I understand this is important, but it raises the question of whether there is value to including all 5 accident and injury categories.

9. Line 288, while it becomes clear that these findings are unadjusted, I recommend labeling them as unadjusted to avoid any confusion.

10. It would help if the authors provided power analyses to assist the reader in understanding the risk of type II error. I see in HRs >2.0 that were not significant. This appears to be a major limitation of the study.

11. The authors did a good job describing other limitations in the manuscript. Some of these pose some risk to the results.

Minor Recommendations:

A. The Abstract seems to set up post-deployment risky behavior as a key topic, but I do not think that is the focus of the manuscript. I recommend revision of the Purpose section.

B. In the Introduction, can you clarify if the stats comparing the Armed Forces to the UK population were controlled for demographic differences?

C. I recommend adding the websites for government studies (e.,g references 1 & 2)

D. In the Demographics section (Line 151), I recommend describing the educational categories for the broad readership of the journal.

E. Center is misspelled for the National Center for PTSD (Line 162)

F. The AUDIT acronym is not defined

G. Line 286 – compared to what group?

H. Line 348 - I do not think military leaders would agree that service members are encouraged to take risks during deployment. Safety is a major focus. I recommend clarifying or deleting.

6. PLOS authors have the option to publish the peer review history of their article (what does this mean?). If published, this will include your full peer review and any attached files.

Reviewer #1: No

---

## [Author Response · Author response to Decision Letter 0]

12 Sep 2022

We have submitted a revised manuscript and responded to reviewers in the attached documents, as requested by journal staff.

---

## [Decision Letter · Decision Letter 1]

24 Oct 2022

PONE-D-21-32763R1Mental health problems and admissions to hospital for accidents and injuries in the UK military: A data linkage studyPLOS ONE

Dear Dr. Chui,

Thank you for submitting your manuscript to PLOS ONE. After careful consideration, we feel that it has merit but does not fully meet PLOS ONE’s publication criteria as it currently stands. Therefore, we invite you to submit a revised version of the manuscript that addresses the points raised during the review process.

We look forward to receiving your revised manuscript.

Kind regards,

Supat Chupradit, Ph.D., M.Ed., B.Sc.(OT), B.P.A., B.Ed., B.A.

Academic Editor

PLOS ONE

Journal Requirements:

Additional Editor Comments:

4 reviewers Accept, but 1 Reviewer Major Revisions.

Please check 1 reviewer that comments and revisions next round.

Reviewers' comments:

Reviewer's Responses to Questions

**Comments to the Author**

1. If the authors have adequately addressed your comments raised in a previous round of review and you feel that this manuscript is now acceptable for publication, you may indicate that here to bypass the “Comments to the Author” section, enter your conflict of interest statement in the “Confidential to Editor” section, and submit your "Accept" recommendation.

Reviewer #1: (No Response)

Reviewer #2: All comments have been addressed

Reviewer #3: (No Response)

Reviewer #4: All comments have been addressed

Reviewer #5: All comments have been addressed

2. Is the manuscript technically sound, and do the data support the conclusions?

Reviewer #1: Partly

Reviewer #2: Yes

Reviewer #3: Yes

Reviewer #4: Yes

Reviewer #5: Yes

3. Has the statistical analysis been performed appropriately and rigorously? 

Reviewer #1: Yes

Reviewer #2: No

Reviewer #3: Yes

Reviewer #4: Yes

Reviewer #5: Yes

4. Have the authors made all data underlying the findings in their manuscript fully available?

Reviewer #1: Yes

Reviewer #2: Yes

Reviewer #3: Yes

Reviewer #4: Yes

Reviewer #5: Yes

5. Is the manuscript presented in an intelligible fashion and written in standard English?

Reviewer #1: Yes

Reviewer #2: Yes

Reviewer #3: Yes

Reviewer #4: Yes

Reviewer #5: Yes

6. Review Comments to the Author

Reviewer #1: The strengths of the paper remain. The authors’ revision was moderately responsive to the feedback. Therefore, the manuscript is improved. However, I have a few remaining recommendations. (Numeric references lines in the manuscript all refer to the Tracked changes version of the document):

1.The “spirit” of some of my prior recommendations related to a sense throughout the paper that the findings are overstated and overemphasized in some places. I still think that needs to be addressed, so here are some clarifications on prior recommendations and some more specific examples:

a.I appreciate the authors adding the limitation in lines 413 about the small cell size. However, their statement that this is a limitation of administrative data from health records is not correct in my opinion. This is a limitation of the rarity of some outcomes and the size of the cohort available for study.

b.I appreciated the authors adding a limitation in Lines 421-424 about including a range of different outcomes. That did not address my original concern, however. I am wondering if there is a different pattern of results for self-inflicted injuries? Given the goals of the paper, should self-inflicted injuries be included with the other accidents and such in this way?

c.I appreciate that the authors do not want to report post-hoc power analyses, but their response did nothing to address a major limitation of the manuscript. Some analyses are not able to detect associations unless they are massive. Some of the CIs are massive. This limitation is not sufficiently addressed.

d.Nearly all the key findings are reporting SMALL effect sizes. This should be emphasized in the abstract, in the initial key findings section of the Discussion, and the conclusion.

e.A thorough read to reduce language that hints at causation is needed. For example, lines 370-371; Line 376 states military personnel are not protected from accidents – the study did not asses this; Line 384-385; Line 386-387 say smoking may play an important role – that is not my read of the results and effect sizes; Lines 389-381 is a bit blunt and overstated; etc. I recommend a critical read by some of the authors who are sensitive to language appropriate for use in association studies.

f.Related to point b, above, the word “may” is used throughout the Discussion to temper interpretations. That is needed but given some of the points that are made and the limitations of the study, I think more clear language is needed. For example, in some places, the authors should be explicit that they are speculating based evidence from prior research. Such language would provide a more appropriate context for the discussion.

g.Additional limitations should be noted:

i.The outcomes were only tracked through 2014. The data are fairly old and changes since then are possible.

ii.The variables used are SO BROAD. Hard to know what some results mean.

iii.The study methods are quite complex. The epidemiological implications and limitations of so many methodological decisions are unknown. The author team has experts in these areas – more epidemiological limitations should be highlighted in my opinion

I regret that I am now including a few new comments below, and I want to be transparent about that since this practice should generally be avoided in my opinion. I defer to the Editor on whether they should be considered:

1.In the Methods where the linkage to health care records is described, it would be helpful to provide some assurances about how comprehensive these records are for the countries named. I confess this may only be a problem for non-UK readers, but in some countries, any attempt to link to health care records would include massive gaps and holes for a population study. If this includes ALL medical care in those countries, that would be a strength to describe in more detail.

2.The Discussion on intentional self-harm (lines 399-406) currently describes the unadjusted analyses as if they are the primary results. This is probably because they are in the expected direction. However, there is only a brief mention of the surprising results that the adjusted analyses were not significant… I recommend reversing the tone of the paragraph to emphasize the adjusted analyses and offering some possible explanations if they exist (e.g., power, something in the methods, something unique about the study topic that could lead to these unexpected results?)

3.Why weren’t participants who only took part at phase 1 included? Was it a consent issue? I recommend specifying (Line 128).

4.I acknowledge this is probably too large of an issue to mention as a new topic in a second review, but in case it is helpful to the authors, I will mention that many readers (maybe most) will wonder whether there were different results for PTSD vs. CMD. They are VERY different. If there is a way to address this, even descriptively, I think readers would appreciate it.

Minor Recommendations:

-Review and remove passive voice

-Avoid the word “dearth,” since it is so overused.

Reviewer #2: 1-this study use multi scales and difference score range. Thus, the authors should show the standard score and criteria in any scale of this study.

2-the data-based that is used in this study should show how select or screen before used in this study. Moreover, the authors should explain how to validate the data for this study.

3-some scales are binary and some scales are interval. The authors should check the data distribution before using logistic analysis.

4-well discussion.

Reviewer #3: An interesting article. Overall, this is a clear, concise, and well-written manuscript. The introduction is relevant, and theory based. Sufficient information about the previous study findings is presented for readers to follow the present study rationale and procedures.

Reviewer #4: All comments have been addressed, Manuscript title: Mental health problems and admissions to hospital for accidents and injuries in the UK military: A data linkage study. I satisfy revision version. Accept

Reviewer #5: (No Response)

7. PLOS authors have the option to publish the peer review history of their article (what does this mean?). If published, this will include your full peer review and any attached files.

Reviewer #1: No

Reviewer #2: No

Reviewer #3: No

Reviewer #4: No

Reviewer #5: No

---

## [Author Response · Author response to Decision Letter 1]

29 Dec 2022

Please see Response to Reviewers attached.

---

## [Decision Letter · Decision Letter 2]

12 Jan 2023

Mental health problems and admissions to hospital for accidents and injuries in the UK military: A data linkage study

PONE-D-21-32763R2

Dear Dr. Chui,

We’re pleased to inform you that your manuscript has been judged scientifically suitable for publication and will be formally accepted for publication once it meets all outstanding technical requirements.

Kind regards,

Supat Chupradit, Ph.D., M.Ed., B.Sc.(OT), B.P.A., B.Ed., B.A.

Academic Editor

PLOS ONE

Additional Editor Comments (optional):

Reviewers' comments:

Reviewer's Responses to Questions

**Comments to the Author**

1. If the authors have adequately addressed your comments raised in a previous round of review and you feel that this manuscript is now acceptable for publication, you may indicate that here to bypass the “Comments to the Author” section, enter your conflict of interest statement in the “Confidential to Editor” section, and submit your "Accept" recommendation.

Reviewer #1: All comments have been addressed

Reviewer #2: All comments have been addressed

2. Is the manuscript technically sound, and do the data support the conclusions?

Reviewer #1: Yes

Reviewer #2: Yes

3. Has the statistical analysis been performed appropriately and rigorously? 

Reviewer #1: Yes

Reviewer #2: Yes

4. Have the authors made all data underlying the findings in their manuscript fully available?

Reviewer #1: Yes

Reviewer #2: Yes

5. Is the manuscript presented in an intelligible fashion and written in standard English?

Reviewer #1: Yes

Reviewer #2: Yes

6. Review Comments to the Author

Reviewer #1: (No Response)

Reviewer #2: well organied and well revise article but some varialbles that were found asscocieted between variables the authors should more clear dissussion.

7. PLOS authors have the option to publish the peer review history of their article (what does this mean?). If published, this will include your full peer review and any attached files.

Reviewer #1: No

Reviewer #2: No

---

## [Editor Report · Acceptance letter]

16 Jan 2023

PONE-D-21-32763R2 

Mental health problems and admissions to hospital for accidents and injuries in the UK military: A data linkage study 

Dear Dr. Chui:

I'm pleased to inform you that your manuscript has been deemed suitable for publication in PLOS ONE. Congratulations! Your manuscript is now with our production department. 

Kind regards, 

on behalf of

Assistant Professor Supat Chupradit 

Academic Editor

PLOS ONE